METHODS

# DeepPL: A deep-learning-based tool for the prediction of bacteriophage lifecycle

**Yujie Zhang[1], Mark Mao[2], Robert Zhang[2], Yen-Te Liao[1], Vivian C. H. Wu[1]***

**1** Produce Safety and Microbiology Research Unit, U.S. Department of Agriculture, Agricultural Research Service, Western Regional Research Center, Albany, California, United States of America, **2** Clowit, LLC. Burlingame, California, United States of America

* vivian.wu@usda.gov

**Data Availability Statement:** All data and codes used for this study are available online. The source code of DeepPL is available via https://github.com/Wu-Microbiology/DeepPL. The model can be downloaded at https://figshare.com/articles/software/DeepPL_model/27005053?file=

## Abstract

Bacteriophages (phages) are viruses that infect bacteria and can be classified into two different lifecycles. Virulent phages (or lytic phages) have a lytic cycle that can lyse the bacteria host after their infection. Temperate phages (or lysogenic phages) can integrate their phage genomes into bacterial chromosomes and replicate with bacterial hosts via the lysogenic cycle. Identifying phage lifecycles is a crucial step in developing suitable applications for phages. Compared to the complicated traditional biological experiments, several tools have been designed for predicting phage lifecycle using different algorithms, such as random forest (RF), linear support-vector classifier (SVC), and convolutional neural network (CNN). In this study, we developed a natural language processing (NLP)-based tool—DeepPL—for predicting phage lifecycles via nucleotide sequences. The test results showed that our DeepPL had an accuracy of 94.65% with a sensitivity of 92.24% and a specificity of 95.91%. Moreover, DeepPL had 100% accuracy in lifecycle prediction on the phages we isolated and biologically verified previously in the lab. Additionally, a mock phage community metagenomic dataset was used to test the potential usage of DeepPL in viral metagenomic research. DeepPL displayed a 100% accuracy for individual phage complete genomes and high accuracies ranging from 71.14% to 100% on phage contigs produced by various next-generation sequencing technologies. Overall, our study indicates that DeepPL has a reliable performance on phage lifecycle prediction using the most fundamental nucleotide sequences and can be applied to future phage and metagenomic research.

## Author summary

Bacteriophages are viruses that infect bacteria and play a critical role in the microbial community within different environments via phage-bacterial evolutionary interactions. The classification of phage lifecycles is of great importance in deploying the potential applications of phages and better understanding complex microbial interactions. However, the traditional biological methods for phage lifecycle identification are complicated and time-consuming. In this study, we proposed a deep learning-based tool—DeepPL—for predicting phage lifecycles using the phage nucleotide genome. Compared with other

49153420. The detailed dataset information, including the NCBI accession number of the phage sequences, verified lifecycle, the usage of training or testing, and the group number used for 5-fold cross-validation, was provided in S1 Data.

**Funding:** This work was supported by the U.S. Department of Agriculture-Agricultural Research Service Current Research Information System projects (2030-42000-055-000-D to YZ, YL, and VCHW). The funders had no role in study design, data collection and analysis, decision to publish, or preparation of the manuscript.

**Competing interests:** The authors have declared that no competing interests exist.

bioinformatic tools, DeepPL was developed from the pre-trained transformers model—DNABERT—designed for fundamental nucleotide language combined with representative phage complete genomes. Our in-house biological results were further used to verify the output of DeepPL. Overall, DeepPL performs with high precision for phage lifecycle prediction and could contribute to the genomic data-driven direction of phage research and applications.

## Introduction

Bacteriophages (or phages) are viruses that infect bacteria and are widely prevalent in different environments, such as oceans, lakes, and agricultural soil, with an estimated number of $10^{31}$ virions in the biosphere [1,2]. There is an increasing number of literature and studies investigating the microbial component from different samples, such as the gut microbiome of humans and farm animals, indicating the vital role of the bacteriophage in the microbiome [3,4]. Phages can influence microbial populations by infecting specific bacterial hosts through two different lifecycles—lytic cycle and lysogenic cycle—based on their nature. Virulent phages (or strictly lytic phages) enter the lytic cycle that utilizes machinery from bacterial hosts to replicate and produce new virions before lysing the host cell. Temperate phages (or lysogenic phages) enter the lysogenic cycle by integrating their DNA into the host genome as prophages (also known as lysogenization). The prophages can be further transmitted to daughter cells at each subsequent bacterial cell division [5]. Under certain environmental stresses, prophages can converse to the lytic cycle by carefully being excised or induced from the bacterial genome to proliferate new infectious phage particles [6].

While lytic phages can infect and promptly lyse bacteria, they are promising antimicrobial agents against antibiotic-resistant bacterial strains. Therefore, lytic phage-based applications have gained attention in different areas, such as phage therapy in clinics and phage-based biocontrol in agriculture [7,8]. For example, a U.S. patient was infected by a superbug of *Acinetobacter baumannii*, which was resistant to all antibiotics. After treating with a lytic phage cocktail, the patient overcame this superbug infection and recovered [7]. However, the lysogenization capability enables temperate phages to be used for several applications, including phage display, genetic manipulation, and pathogen detection [9–11]. Phage display has been applied in drug discovery and antibody production to treat human diseases [12].

Identifying the lifecycle of newly isolated phages is a critical step highly associated with subsequent applications. Previously, plaque morphology via plaque assay was used as a traditional biological method to identify phage lifecycle: virulent phages produce clear plaques, and temperate phages produce turbid plaques [13]. Due to the inaccurate results from plaque morphology identification, several methods have been developed for isolating and screening the potential lysogens, such as Digoxigenin (DIG)-labeling in situ hybridization and patch test [14,15]. Though these methods improve accuracy and sensitivity, they are costly and time-consuming. Taking advantage of next-generation sequencing (NGS) development, the whole genome sequencing of isolated phages becomes helpful in determining the phage lifecycle by genomic features. The most common strategy is to compare new phage sequences with the reference phage genomes, which have known lifecycles, and detect the presence of lysogenic genetic modules through basic local alignment search tools (BLAST). In recent years, machine learning (ML) has been utilized to develop tools for classifying phage lifecycles into two categories based on the input file types (nucleotide sequence or amino acid sequences of complete phage genomes). The first ML-based tool—PHACTS—was developed to categorize phage

lifecycles using the amino acid sequences from the annotated phage genome via a random forest algorithm [16]. This tool was published in 2012; however, its accuracy dropped due to the lack of subsequent maintenance. In 2020, Tynecki et al. proposed a novel tool called phageAI, which was able to classify phage lifecycles via the combination of the Word2Vec Skip-gram model and a linear support-vector classifier (SVC) with an average accuracy of 98.90% on the small size of validation sets [17]. Deephage, developed by Wu et al., used a "one-hot" encoding and a convolutional neural network (CNN) to predict phage lifecycles, but it only achieved the best performance of 89% [18].

Bidirectional Encoder Representations from Transformers (BERT) has been widely used in natural language processing (NLP). It is composed of two steps: pre-training and fine-tuning. Notably, the BERT model was pre-trained in many different NLP tasks. Fine-tuning with a relatively small training set can yield great results for new downstream tasks. This distinctive feature of the unified architecture across various downstream tasks demonstrates its potential applications in different areas, including the biological language. Most recently, a newly published tool—PhaTYP—was designed to predict the lifecycles of phages using protein sequences via BERT, showing 98% accuracy [19]. However, protein-based (amino acid) sentences heavily rely on the well-known biological function of the proteins related to the lysogenic cycle, while most newly identified phage sequences do not have known counterparts. Nucleotide sequences (also called DNA sequences), constructed by assembling the four nucleotides (A, T, G, C base), are the most fundamental biological elements and contain all genetic information necessary for encoding functional molecules, including protein [20,21]. Moreover, compared to the BERT, DNABERT has been developed by training BERT using human DNA genomes [22]. Taking advantage of DNABERT with a better understanding of genomic DNA sequences, in this study, we developed a DNABERT-based tool, DeepPL, to identify the phage lifecycle with the input of phage nucleotide sequences (Fig 1). The improvement from amino acid sequences to nucleotide sequences overcame the missing details from translation and imperfect annotation of phage protein but also explored the fundamental instructions of non-coding DNA language related to the phage lifecycle. The pre-trained transformers model designed for DNA language combined with representative phage complete genomes will significantly enhance the accuracy of phage lifecycle classification and further contribute to the data-driven direction of phage-based research and application.

## Materials and methods

### Data collection

For the training dataset, the datasets from Deephage were collected and further subjected to the following trimming process [18]. First, the downloaded phage sequences were manually curated with reliable lifecycle annotations through the literature review. Second, based on the nature of virulent and temperate phages, the key genetic difference between these two phage types is the lysogenic genes within the temperate phage genomes. Therefore, several representative lysogenic gene markers for temperate phages were selected, including integrase, excisionase, recombinase, regulatory protein cro, antitermination protein Q, cI repressor, cII protein, cIII protein replication protein O, replication protein P, and recombination protein Bet. Third, we extracted these lysogenic genes from the phage genomes prior to manually correcting and removing the incorrect data deriving from their previous gene annotation. Finally, we combined the nucleotide sequences of each selected gene to generate a lysogenic dataset for our current study. After the above processes, we obtained a benchmark dataset with 1,488 lysogenic genes from 557 temperate phage genomes, which were further used for model training. Compared to temperate phages, more virulent phages with biological verification have been

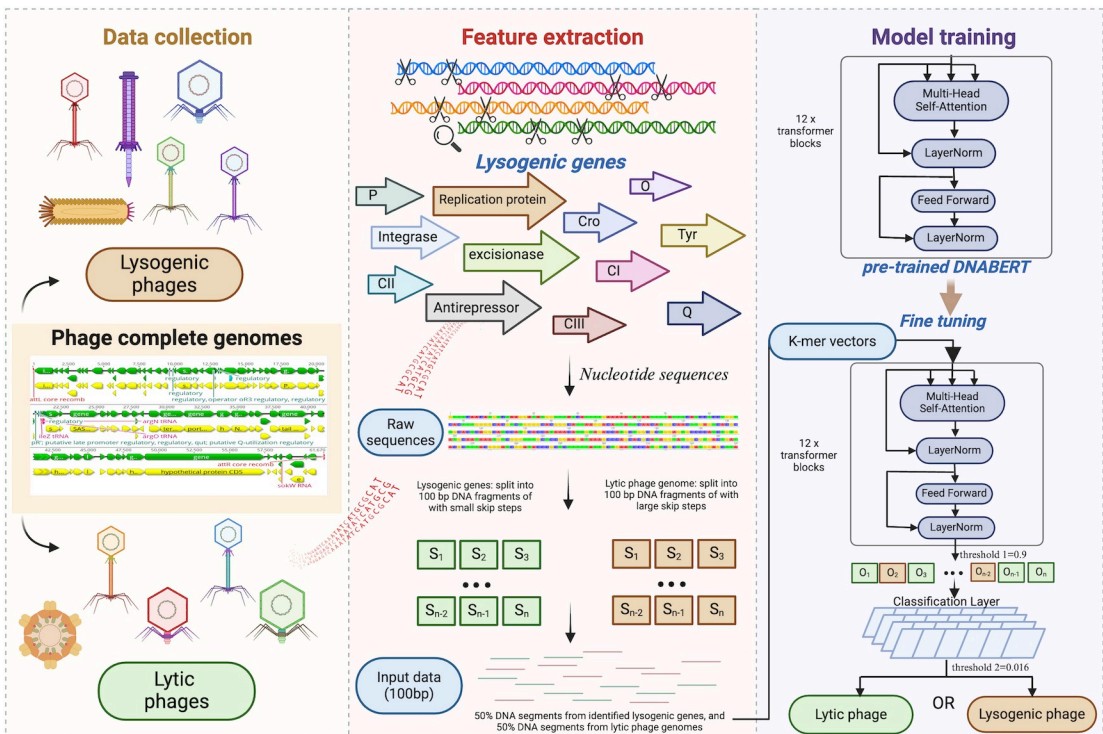

**Fig 1. An overview of DeepPL framework for predicting phage lifecycle, including data collection, feature extraction, and model training.** Diverse lysogenic and lytic complete phage genomes were collected from the National Center for Biotechnology Information (NCBI) database. Non-ATGC letters within the phage nucleotide sequences were randomly replaced by ATGC letters. The phage lifecycles were manually confirmed by the literature review. Further, the lysogenic genes were identified and extracted from lysogenic phage genomes. The sliding window of 100 bp in length and further conversion of sets of k-mer 6 sequences from phage sequences were used as input for a fine-tuning training process based on the pre-trained DNABERT model. The process generated the binary classification probability (0–1) of each k-mer 6 sequence. Therefore, threshold 1 was used to identify a good match between phage sequences and lysogenic genes. The threshold 1 of the binary classification probability above 0.9 was identified as a good match between 100 bp DNA segments and lysogenic genes. Further, the results from the frame-by-frame classification results were aggregated into one final classification result for phage lifecycle prediction with the threshold 2 of 0.016. The input phage sequence with a threshold 2 below 0.016 was identified as a lytic phage; otherwise, it was predicted as a lysogenic phage. The figure was created using BioRender. Zhang, Y. (2024) BioRender.com/w89b419.

reported in the National Center for Biotechnology Information (NCBI) database. Therefore, a total of 1,262 virulent phage genomes with the strictly lytic cycle were verified and used to balance the training dataset. To test the performance of our model and compare it with other state-of-the-art tools, an additional 374 complete phage genomes with reliable lifecycle labels were collected from the NCBI database as the test dataset, including 245 virulent phages and 129 temperate phages.

All the complete phage genomes were downloaded from the NCBI database. Detailed information on each phage genome was provided in S1 Data, including the phage accession number, verified lifecycle, the usage of training or testing, and the group number used for 5-fold cross-validation.

## Model training and testing

Taking advantage of DNABERT with trained knowledge of genome sequences at the nucleotide level, we started with the DNABERT model and used phage nucleotide sequences to do the fine-tuning training process for our specific classification task. Specifically, the phage nucleotide sequences have been pre-checked for the presence of non-ATGC letters for noise

reduction. The sequences of less than 10 non-ATGC letters were replaced with random letters of ATGC. The sequences of more than 10 non-ATGC letters were removed from the training dataset. To classify the phage lifecycle (lysogenic cycle vs. lytic cycle), a balanced training set, consisting of 50% nucleotide sequences from identified lysogenic genes and 50% nucleotide sequences from lytic phage genomes, was obtained by sliding window-based selection and further employed for model training. K-mer 6 was selected in this study based on its best performance, as reported by DNABERT. The sequence sizes between 100 to 512 bp were compared in the fine-tuning process. The comparison of different parameters indicated that the sequence size of 100 bp yielded reasonably balanced results with less computing requirements and was further used in the current study. To that end, the input of training sequences was obtained through a sliding window of 100 bp in length and turned into a set of k-mer 6 sequences. In addition, the lysogenic genes extracted from lysogenic phage genomes are much smaller than the complete genomes of lytic phages. Therefore, different skip steps were selected for lysogenic genes and lytic phage genomes to get a 50%-50% balanced training dataset (547,810 DNA sequences from lytic phage genome and 500,765 DNA sequences from lysogenic genes) with the full coverage of lysogenic genes in this study. Specifically, a small skip step of 1 (i.e., genome location, base pairs 1–100, 2–101, 2–102, etc.) was used for the sliding window in the lysogenic gene to cover limited lysogenic features. Further, a large skip step of 91 (i.e., genome location, base pairs 1–100, 92–191, 183–282, etc.) was selected for the sliding window in the lytic phage genomes to match the number of DNA sequences from lysogenic genes. After evaluation, around 10 epochs of training were conducted during the fine-tuning process, with a larger learning rate in early epochs and a smaller learning rate in later epochs.

The same setting of a sliding window of 100-bp sequence size and the subsequent conversion of k-mer 6 sequence sets was for the model testing. In detail, the entire set of k-mer 6 sequences was run using our model to generate the binary classification probability (0–1) of each k-mer 6 sequence. By optimizing the threshold values, high sensitivity and accuracy were achieved and subsequently used in this study based on the following settings: threshold 1 of 0.9 was set for good matching between each 100 bp segment and lysogenic genes, and threshold 2 of 0.016 was used to aggregate the frame-by-frame classification results into one final classification result for phage lifecycle prediction. The phages with a probability below 0.016 were predicted as lytic phages, whereas phages with a probability equal to or above 0.016 were predicted as lysogenic phages.

## Performance evaluation

The test dataset was used to evaluate the performance of DeepPL and compare it with other tools for phage lifecycle prediction. A total of five performance measures, including sensitivity (SN), specificity (SP), accuracy (ACC), F-score, and MCC, were selected to quantify the performance in this study [23]. For evaluation, temperate and virulent phages are referred to as positive and negative samples, respectively.

## Results

### Performance assessment

The 5-fold cross-validation was conducted to evaluate the performance stability of DeepPL by randomly dividing the training dataset (1,262 virulent phage genomes and 1,488 lysogenic genes from 557 temperate phage genomes) into five equal-sized groups (around 450,000 DNA sequences) (S1 Data). Each group contained approximately 360 phage sequences, consisting of 50% of the lysogenic gene sequences (i.e., around 229,055 sequences in Group 1) extracted from lysogenic phage genomes and 50% of DNA fragments (i.e., around 226,626 sequences in

**Table 1. The performance comparison between DeepPL and previously published tools for phage lifecycle prediction.**

| Tools | Sensitivity (%) | Specificity (%) | Accuracy (%) | F-score | MCC |
|---|---|---|---|---|---|
| DeepPL | **92.24** | 95.91 | 94.65 | **0.92** | **0.53** |
| PhaTYP | 90.44 | 97.47 | **94.91** | **0.92** | **0.53** |
| Deephage | 78.61 | **98.13** | 89.83 | 0.86 | 0.49 |
| PHACTs | 38.94 | 79.77 | 48.66 | 0.53 | 0.14 |
| PhageAI | 83.33 | 96.08 | 91.17 | 0.87 | 0.50 |

Group 1) extracted from lytic phage genomes. Specifically, each group (i.e., Group 1) was used as the test dataset, while the rest of the groups (i.e., Groups 2–5) were used as training datasets. The result showed that the sensitivity of the five groups was from 85% to 96%, while the specificity of each group ranged from 89% to 94% (S2 Data). Moreover, the accuracy from 90.6% to 93.9% indicated that DeepPL has an overall reliable and stable performance for phage lifecycle prediction.

## Comparison with previously published methods

To estimate the predictive capability of DeepPL, four current available tools designed for phage lifecycle prediction, including PhaTYP, Deephage, PhageAI, and PHACTs, were selected to compare the performance with DeepPL in this study using the test dataset (Table 1). The performance comparison revealed that our DeepPL and PhaTYP yielded better results than the other three tools, suggesting the advantage of NLP applied in the genomic language. DeepPL had a slightly higher sensitivity than PhaTYP, while the specificity of PhaTYP was slightly better than that of DeepPL. The accuracy of DeepPL and PhaTYP showed the best results, around 94%, and outperformed the other tools. Most importantly, compared to the amino-acid-based PhaTYP, DeepPL can achieve a similar performance as PhaTYP using DNA sequences. In addition, the ROC curve comparison on the test dataset was also derived to illustrate the trade-offs between the sensitivity and specificity of each tool (Fig 2). The AUCROC score showed DeepPL had the best performance (0.98), followed by DeePhage (0.97) and PhaTYP (0.85). Together, the results demonstrated that DeepPL could capture the underlying

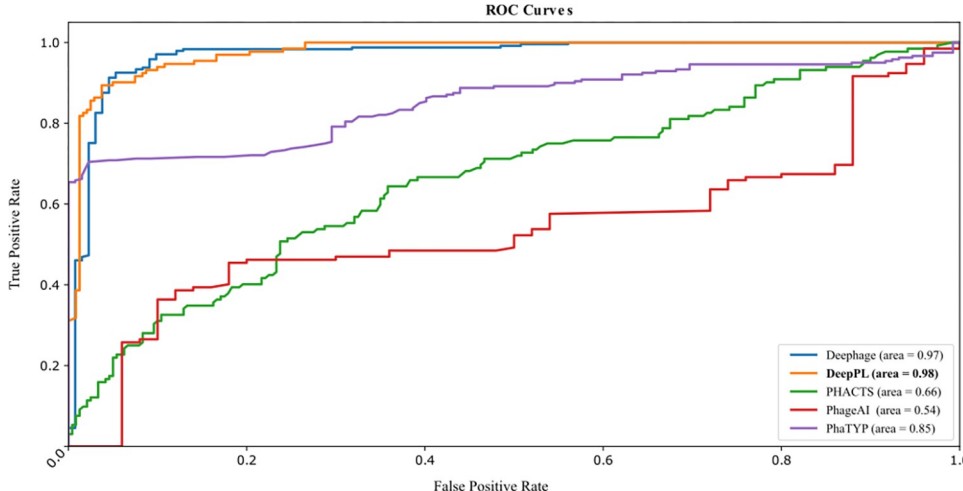

**Fig 2. The ROC curve comparison between DeepPL and previously published tools for phage lifecycle prediction on test dataset.** The value shown in the legend is AUCROC score.

genomic differences between virulent and temperate phage genomes at the fundamental nucleotide level without trading off the accuracy.

## Case study 1-In-house verified phages

In this study, 19 genomes from the phages isolated by our lab were used via DeepPL prediction to validate the prediction results and evaluate the potential application in phage studies (Table 2). Biological and genomic characterizations were used to confirm the lifecycles of 19 phages isolated from different sources. The biological experiments included the spot test, plaque assay, bacterial challenge assay, and lysogen test. The genomic analysis included whole-genome sequencing, lysogenic gene detection, and comparative genomics with the published reference phages and bacterial host. Among these phages, only the phage Sa179lw has a questionable lifecycle. In detail, this phage showed lytic activity against *E. coli* O179 strains via biological tests but was detected with the presence of the lysogenic module by the genomic analysis. The results showed that DeepPL had 100% accuracy in predicting the lifecycles of the 18 in-house verified phages. Moreover, the lysogenic potential of the questionable phage Sa179lw was detected by DeepPL but not by PhaTYP, due to the high sensitivity of DeepPL. Overall, the prediction results demonstrated that our DeepPL can be a promising tool for phage research and phage-based applications.

## Case study 2-Metagenomic data

Viral metagenomics has been developed to understand viral diversity and ecology within diverse environments, such as mammalian gastrointestinal (GIT) and agricultural environments. In this case, DeepPL was employed to evaluate its potential application for phage

**Table 2. The performance validation of DeepPL on phage lifecycle prediction using in-house verified phages.**

| Phages | Accession number | Sequence length (bp) | DeepPL prediction | PhaTYP prediction | Confirmed lifecycle | References |
|---|---|---|---|---|---|---|
| *Escherichia* phage vB_EcoP-Ro103C3lw | MN067430 | 39,389 | Lytic | Lytic | Virulent | [31] |
| *Escherichia* phage Ro45lw | MK301532 | 39,793 | Lytic | Lytic | Virulent | [32] |
| *Escherichia* phage vB_EcoS-Ro145clw | MG852086 | 42,031 | Lytic | Lytic | Virulent | [33] |
| *Salmonella* phage S4lw | OQ660438 | 42,250 | Lytic | Lytic | Virulent | NA |
| *Escherichia* phage vB_EcoS-UDF157lw | OQ243221.1 | 46,604 | Lytic | Lytic | Virulent | NA |
| *Escherichia* phage Lys8385Vzw | MT225100 | 50,953 | Lysogenic | Lysogenic | Temperate | [34] |
| *Escherichia* phage Lys19259Vzw | MT225101 | 61,072 | Lysogenic | Lysogenic | Temperate | [34] |
| *Escherichia* phage Lys12581Vzw | NC_049917 | 62,668 | Lysogenic | Lysogenic | Temperate | [35] |
| *Escherichia* phage vB_EcoM-Ro157lw | MH051335 | 72,179 | Lytic | Lytic | Virulent | [36] |
| *Escherichia* phage vB_EcoM-Ro111lw | MH571750 | 86,950 | Lytic | Lytic | Virulent | [36] |
| *Escherichia* phage vB_EcoM-Pr103Blw | MW481326 | 88,421 | Lytic | Lytic | Virulent | [31] |
| *Escherichia* phage vB_EcoM-Pr121LW | MH752840 | 134,575 | Lytic | Lytic | Virulent | [37] |
| *Escherichia* phage vB_EcoM-Ro121lw | MH160766 | 149,803 | Lytic | Lytic | Virulent | [36] |
| *Escherichia* phage vB_EcoM_Sa157lw | MH427377 | 155,887 | Lytic | Lytic | Virulent | [8] |
| *Salmonella* phage D5lw | OQ660437 | 157,399 | Lytic | Lytic | Virulent | NA |
| *Escherichia* phage vB_EcoM-S1P5QW | OL956808 | 166,102 | Lytic | Lytic | Virulent | [38] |
| *Escherichia* phage vB_EcoM-G157lw | OK331996 | 167,170 | Lytic | Lytic | Virulent | NA |
| *Escherichia* phage vB_EcoM-Sa45lw | MK977694 | 167,353 | Lytic | Lytic | Virulent | [39] |
| *Escherichia* phage vB_EcoS Sa179lw | MH023293 | 46,833 | Lysogenic | Lytic | Questionable* | [40] |

NA: the information is not available

* The phage has lysogenic potential and its lifecycle hasn't been confirmed by biological experiments.

lifecycle classification in metagenomic research. However, next-generation metagenomic sequencing is still at an early stage in investigating viral communities; numerous phage genomes have been detected in the metagenomic sequencing dataset, but biological experiments could not isolate and verify the phage particles. Therefore, a mock phage community metagenomic dataset was selected to accurately compare the performance among different tools instead of the real viral metagenomic dataset. Specifically, this mock phage community, composed of 15 sequenced phages with known lifecycles, was constructed by Cook et al., to perform metagenomic sequencing using the most common next-generation sequencing technologies, including Illumina, Pacbio, and Nanopore sequencing [24]. Therefore, we downloaded 15 complete phage genomes and the metagenomic sequences of this mock phage community generated by various sequencers to evaluate whether our DeepPL could identify the phage lifecycle correctly using metagenomic data. In addition, the PhaTYP was also employed for accuracy comparison due to the similar performance shown above.

First, the 15 complete phage genomes were predicted for lifecycles by DeepPL and PhaTYP (Table 3). The results showed that DeepPL and PhaTYP had 100% and 75% accuracy in predicting these phages with known lifecycles, respectively, indicating a better prediction using DeepPL than PhaTYP with complete phage genomes. The phage sequences were generated from different sequencing platforms (Illumina, Pacbio, and Nanopore) alone or in combination and further assembled by various software (Unicycler, Spades, Fyle, wtdgb2). The resulting phage contigs sharing high genomic similarity with the 15 phage genomes via the nucleotide basic local alignment search tool (blastn) were further used to determine the performance of DeepPL and PhaTYP (Fig 3). For the contigs assembled by Illumina short reads, DeepPL and PhaTYP achieved 84% and 44% accuracy, respectively. Furthermore, DeepPL had high accuracies ranging from 71.14% to 100% for the prediction of contigs generated by Nanopore and Pacbio long reads, while PhaTYP showed varied accuracies of 49.66% - 100%. Among the contigs co-assembled by Illumina short reads and Nanopore/Pacbio long reads, DeepPL performed a relatively higher accuracy (83.33% - 100%) than PhaTYP (55.55% - 100%). The quality of metagenomic sequencing (i.e., genome recovery and error rates) was

**Table 3. The phages with known lifecycles used to construct a mock phage community for metagenomic sequencing.**

| Phages | Accession number | Sequence length (bp) | DeepPL Prediction | PhaTYP prediction | Confirmed lifecycle | References |
|---|---|---|---|---|---|---|
| phiX174 | NC_001422 | 5,386 | Lytic | Lytic | Virulent | [41] |
| HP1 | NC_001697 | 32,355 | Lysogenic | Lysogenic | Temperate | [42] |
| KUW1 | OQ376857 | 44,509 | Lytic | Lytic | NA | NA |
| PARMAL1 | OQ376858 | 44,565 | Lytic | Lytic | NA | NA |
| J1 | LR027388 | 50,343 | Lysogenic | Lytic | Temperate | [43] |
| J2 | LR027385 | 50,343 | Lysogenic | Lytic | Temperate | [43] |
| SWAN | LT841304 | 50,865 | Lysogenic | Lytic | Temperate | [43] |
| CDMH1 | NC_024144 | 54,279 | Lysogenic | Lysogenic | Temperate | [44] |
| VP1 | NA | 70,044 | Lytic | Lytic | Virulent | [45] |
| SM032 | OV032860 | 79,660 | Lytic | Lytic | Virulent | [45] |
| J3 | LR027389 | 115,471 | Lytic | Lytic | Virulent | [43] |
| DSS3Mal1 | NA | 149,582 | Lytic | Lytic | NA | NA |
| PHAGE1 | LR027390 | 167,773 | Lytic | Lytic | Virulent | [45] |
| S-RSM4 | FM207411 | 194,454 | Lytic | Lytic | Virulent | [46] |
| vB_Vpa_sm033 | OV032902 | 320,253 | Lytic | Lytic | Virulent | [45] |

NA: the information is not available

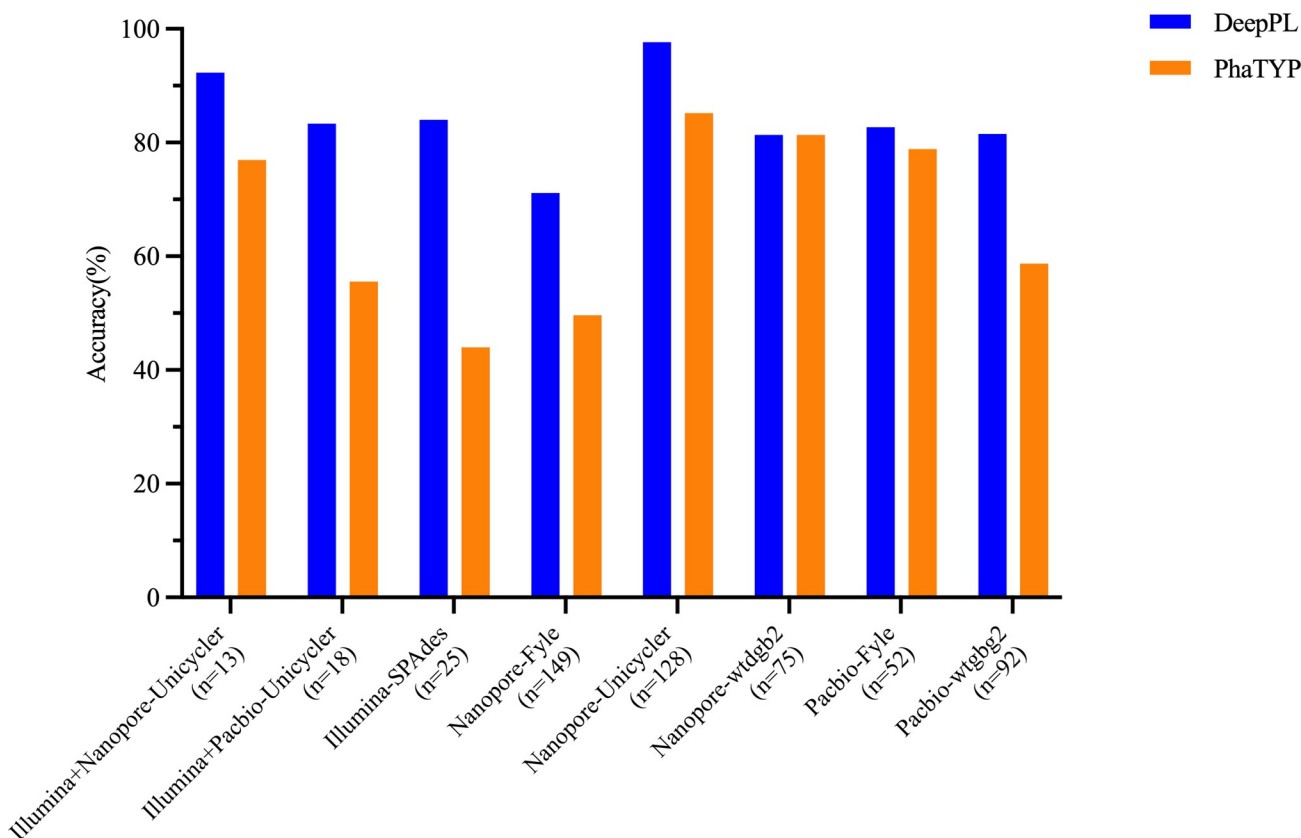

**Fig 3. Comparison of DeepPL and PhaTYP performance on phage contigs in a mock phage community generated by different metagenomic sequencing technologies and assemblies.**

highly correlated to the sequencer and assembler used in the study and further affected the downstream analysis, such as phage lifecycle prediction. Moreover, the contigs generated by short-read and long-read sequencing approaches were much shorter than complete phage genomes. Even so, DeepPL was able to identify the phage features and predict phage lifecycles with higher accuracy using metagenomic sequences, suggesting its potential as a useful bioinformatic tool in metagenomic research.

## Discussion

As the most abundant biological entities in the biosphere, many bacteriophages remain uncharacterized and require further exploration [25]. Diverse phage sequences continue to be discovered through bacteriophage and metagenomic research. But still, a vast majority of available phage sequences create considerable difficulties in navigating through all the data in search of biological meaning, the annotation of phage proteins in particular. Currently, no standard workflow is available for full annotation of a draft phage genome. Due to the imperfect and inconsistent annotation of phage genomes, it is hard to achieve precise characterization based on the phage amino acid sequences. Therefore, nucleotide sequences (DNA sequences) are the most fundamental genomic language that can represent basic biological information accurately. In this study, we utilized the comprehensive language learning ability, DNABERT, to characterize phage lifecycle by feeding the fundamental DNA sequences. It could reduce the noise from protein prediction or rough phage genome annotation from

different pipelines to capture the underlying semantic information of DNA language for phage lifecycle identification.

The phage lifecycle classification requires a complicated validation process through traditional biological experiments, including the spot test, plaque assay, and lysogen test. A precise prediction of the phage lifecycle will improve biological validation tests by saving time and effort. In this study, the precision of DeepPL was also confirmed by the biologically-characterized phages in case study 1. A total of 19 different phages were isolated from agricultural environments by our lab and subjected to verify the phage lifecycles by essential biological experiments. The DeepPL analysis showed the prediction is 100% consistent with our biological results. To our knowledge, this is the first study using actual phages characterized by biological experiments to verify the output of prediction models.

Solid lytic phages can only perform the lytic cycle, whereas lysogenic phages can display either a lytic or lysogenic cycle based on environmental conditions. Therefore, virulent and temperate phages both contained many lysis modules. The primary genomic difference between them relies on the presence of limited lysogenic genes. Therefore, only a few DNA segments within the phage genome can be detected as lysogenic modules. It further explained the reasonable setting of the low threshold 2 that the score of phage classification more than 0.016 was identified as lysogenic phage in this study. In addition, threshold 2 of 0.016 contributes to the high sensitivity of lysogenic gene detection using our model. This capability could benefit phage applications in the clinical and agricultural areas. Specifically, lytic phages used for phage-based therapy or biocontrol have a strict requirement: no virulence, antibiotic resistance, or lysogenic genes can be detected in the phage genome. In nature, some lysogenic phages lose their lysogenic genes during the phage production process and perform the lytic cycle. In some cases, lytic phages could acquire one or a few lysogenic genes by gene exchange or recombination but still show strict lytic infection. Due to the risk of lysogenic potential, these phages will not be considered for phage therapy or biocontrol purposes. For example, one of our lab phages presented in this study, Sa179lw, was isolated from surface water, showing antimicrobial activity against *E. coli* O179 strains. However, this phage was withdrawn from the lytic phage cocktail development for biocontrol application because the lysogenic gene *cro* was present in the genome and also identified by DeepPL. The findings indicate the excellent screening capability of DeepPL on the phenotypic camouflage phages.

Viral metagenomic sequencing has become a popular technique to determine the viral population within different samples, including gut and environmental viromes [26,27]. Bacteriophages, the major component of the virome, drive the diversities and evolution of viral communities and host bacterial populations. Due to the specific interaction between phages and bacteria, there has been a growing interest in lytic and lysogenic phage profiles within the total phageome in recent years. Therefore, the potential application of DeepPL in the metagenomic study was tested in case study 2. DeepPL performed well using short-reads and long-reads-based metagenomic sequences longer than 5,000 bp with accuracy ranging from 82.66% to 100%. Even for sequences less than 5,000 bp, DeepPL has a relatively high accuracy of 71.14% - 97.65%, depending on the sequencer and assembly tools (S3 Data). The results from this study indicated that the accuracy was closely associated with several factors, including the contig lengths, sequencing technologies, and genome assembly tools. Notably, the short contigs require a comprehensive capability to identify genomic features of lysogenic genes for classification. As a result, PhaTYP's prediction accuracy decreased in predicting contigs shorter than 5,000 bp, likely due to limited amino acid sequences by genomic annotation. In contrast, with a relatively high accuracy, the DeepPL model can efficiently identify the lysogenic gene feature based on the nucleotide sequences.

DeepPL has a great performance on phage lifecycle classification compared to currently available tools. Even so, some limitations still need to be improved in future studies. The precision of DeepPL trades off the rapid processing time. The processing time of DeepPL depends on the length of phage genomes and the processor. For example, the CPU (M1 chip) can predict a phage genome with a length of 42,031 bp in 17 mins, while the GPU (NVIDIA V100 GPU, 1 node, memory of 64 GB) can complete the prediction within 2 mins. In addition, a user-friendly web server version of DeepPL will be established for users unfamiliar with coding languages. Furthermore, the diversity and plasticity of phage genomes are highly associated with the model performance. The predictive ability of phylogeny on diverse microbial populations has been shown in several studies [28–30]. Therefore, the phylogenetic-based sampling method, with more newly sequenced phage genomes, should be included in the training dataset to increase the coverage of DeepPL.

## Supporting information

**S1 Data. Detailed phage information of training and test datasets.**
(XLSX)

**S2 Data. The performance of DeepPL using 5-fold cross-validation.**
(XLSX)

**S3 Data. Detailed prediction results of DeepPL and PhaTYP for case-study2.**
(XLSX)

## Author Contributions

**Conceptualization:** Vivian C. H. Wu.

**Data curation:** Yujie Zhang.

**Funding acquisition:** Vivian C. H. Wu.

**Investigation:** Yujie Zhang.

**Methodology:** Yujie Zhang, Mark Mao, Robert Zhang.

**Software:** Mark Mao, Robert Zhang.

**Supervision:** Vivian C. H. Wu.

**Writing – original draft:** Yujie Zhang.

**Writing – review & editing:** Yujie Zhang, Mark Mao, Robert Zhang, Yen-Te Liao, Vivian C. H. Wu.

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
