## [Decision Letter · Decision Letter 0]

20 May 2024

Dear Dr. Wu,

Thank you very much for submitting your manuscript "DeepPL: a deep-learning-based tool for the prediction of bacteriophage lifecycle" for consideration at PLOS Computational Biology.

As with all papers reviewed by the journal, your manuscript was reviewed by members of the editorial board and by several independent reviewers. In light of the reviews (below this email), we would like to invite the resubmission of a significantly-revised version that takes into account the reviewers' comments.

We cannot make any decision about publication until we have seen the revised manuscript and your response to the reviewers' comments. Your revised manuscript is also likely to be sent to reviewers for further evaluation.

Sincerely,

Yang Lu, Ph.D.

Academic Editor

PLOS Computational Biology

Stacey Finley

Section Editor

PLOS Computational Biology

Reviewer's Responses to Questions

**Comments to the Authors:**

Reviewer #1: The authors introduce a deep learning tool based on natural language processing techniques to predict bacteriophage lifestyles from nucleotide sequences (DeepPL). This tool uses the DNABERT model, a variant of the popular BERT model adapted for DNA sequences, and demonstrates an accuracy of 94.65%, with sensitivity and specificity rates of 92.24% and 95.91% respectively. DeepPL performs comparably to existing tools and has shown promising results in both lab-verified phage genomes and metagenomic datasets, achieving 100% accuracy on individual phage genomes and varying high accuracies on phage contigs. However, I have several concerns about this work:

1. Figure 1 Clarification: Figure 1 is difficult to interpret. For instance, in the Feature Extraction panel, it's unclear what the annotations P, O, Q, CI represent-are these proteins? Also, does DeepPL require protein inputs? What does the highlighted ACGT sequence signify? In the Model Training panel, what does "12 X" mean? Is it indicating that the input K-mer vectors are processed through 12 Transformer blocks? Are the pre-training and fine-tuning steps using the same Transformer block structures? He, et al [1] mention the use of residual connections to prevent overfitting. It would be beneficial to include it in the Transformer block layer normalization.

2. Dataset Description: It would be beneficial to include a brief description of the datasets in the Results section for better clarify and context.

3. Performance Assessment: The rationale behind using a balanced training set in the 5-fold cross-validation should be explained. Additionally, Table 1 seems redundant and could potentially be omitted for brevity.

4. Method Comparison: The comparison with previously published methods should include a ROC curve to better illustrate the trade-offs between sensitivity and specificity. Also, it would be good if author could explain the low sensitivity observed with PHACTs. The running time comparison of different methods should be included.

5. Case Study 1: Both PhaTYP and DeepPL predict with 100% accuracy for the lifestyle of 18 phages, yet one questionable phage, Sa1791w, presents a controversial lifestyle, making the conclusion that PhaTYP has lower accuracy than DeepPL. The conclusion is unconvincing.

6. Case Study 2: Clarify what “blastn” refers to in the text. Additionally, explain the presence of several NAs in Table 4 and ensure consistency in the formatting of the ‘Sequence length (bp)’ column across tables. The sequencing platform and assembly method should also be included. For Figure 2, include the sample size for each bar and consider using blue/red for better visual distinction as complementary colors.

7. Data Collection Consistency: The authors state that representative lysogenic gene markers were selected, but performance on uncoded genes or a mix (partial coded) would be interesting to see. Otherwise, I feel like no difference between PhaTYP and DeepPL. Discrepancies in the number of lytic and lysogenic samples between the manuscript and Supplementary Data 2 should be addressed.

8. K-mer, sliding window and Learning Rate: Clarify what “k-mer 6 sequences” refers to – is it sequences of six nucleotides? The decision to use a skip step for a balanced training dataset, use 100 bp in sliding window and the relationship between learning rate and epoch should be elaborated on.

9. Minor. At line 111, could you please clarify whether you are referring to the conversion from amino acid sequences to nucleotide sequences?

10. Minor. In the discussion section, consider including references [2, 3], which discuss the use of phylogenetic tree information (highlighting greater similarity among closely related entities) to improve the predictive power and association test power.

[1] He K, Zhang X, Ren S, et al. Deep residual learning for image recognition. In: Proceedings of the IEEE Conference on Computer Vision and Pattern Recognition, Manhattan, New York, NY, United Stated: Institute of Electrical and Electronics Engineers, 2016, 770–8.

[2] Walkup, J., Dang, C., Mau, R.L. et al. The predictive power of phylogeny on growth rates in soil bacterial communities. ISME COMMUN. 3, 73 (2023). https://doi.org/10.1038/s43705-023-00281-1

[3] Hong, Q., Chen, G. & Tang, ZZ. PhyloMed: a phylogeny-based test of mediation effect in microbiome. Genome Biol 24, 72 (2023). https://doi.org/10.1186/s13059-023-02902-3

Reviewer #2: The authors present a deep-learning tool (DeepPL) based on the previously published DNABERT (Bidirectional Encoder Representations from Transformers for DNA language) model. The goal of DeepPL is to classify phage genomes as either lytic or lysogenic based on their sequence. DeepPL uses the pre-trained DNABERT model and fine-tunes it for the new task in the space of Bacteriophages. The authors demonstrate that their tool performs favorably to four other methods. It is an interesting problem and as we need to improve our understanding of the interactions between viruses and bacteria, I think this work is timely. Overall, I find the paper to be well written, however, I have two major concerns about the way key parameters were selected as discussed below.

Major remarks:

Line 300 “The comparison of different parameters indicated that k mer 6 and sequence size of 100 bp yielded reasonably balanced results and were further used in the current study.“ The comparison is nowhere to be found in the manuscript and no description is given as to what a balanced result is

How did you decide on the size of the skip step for lytic phage genomes? What tests were done to justify a much larger step size of 91? Why should different skip-steps be used to begin with (you can balance the sets at k-mer level rather than gene level)? When you use skip-step=1, you will have many overlapping k-mers, with skip-step=91, none at all. How would this affect the model training and performance? I found the lack of any evidence and discussion to support the choice of (all) your parameters astonishing

You report results based on five fold cross-validation. Yet, in the Data Collection section (line 283) it is stated that “additional 374 complete phage genomes with reliable lifecycle labels were collected from the NCBI database as the test dataset.” So, was it a 5-fold cross-validation or an external testset?

Minor remarks:

It’s unclear in the caption of Figure 1 what threshold 1 is as it is not in the figure.

How was the aggregation of frames done to obtain a score for threshold2?

Line 281: “Therefore, a total of 1,262 virulent phage genomes with the strictly lytic cycle were verified and used to balance the training dataset.” - expand to make clear you balance the number of genes.

Fig2: colors are hard to distinguish

Table3: mark in bold the best performer in each category

Table4: hard to read as column 1 takes two or there rows

Reviewer #3: Uploade as attachment

**Have the authors made all data and (if applicable) computational code underlying the findings in their manuscript fully available?**

Reviewer #1: Yes

Reviewer #2: Yes

Reviewer #3: **No: **The code and the downloadable data are not found in the manuscript.

PLOS authors have the option to publish the peer review history of their article (what does this mean?). If published, this will include your full peer review and any attached files.

Reviewer #1: No

Reviewer #2: **Yes: **Borislav Hristov

Reviewer #3: No
---

## [Decision Letter · Decision Letter 1]

31 Aug 2024

Dear Dr. Wu,

Thank you very much for submitting your manuscript "DeepPL: a deep-learning-based tool for the prediction of bacteriophage lifecycle" for consideration at PLOS Computational Biology. As with all papers reviewed by the journal, your manuscript was reviewed by members of the editorial board and by several independent reviewers. The reviewers appreciated the attention to an important topic. Based on the reviews, we are likely to accept this manuscript for publication, providing that you modify the manuscript according to the review recommendations.

The authors should address the remaining critiques to strengthen the paper.

Sincerely,

Yang Lu, Ph.D.

Academic Editor

PLOS Computational Biology

Stacey Finley, Ph.D.

Section Editor

PLOS Computational Biology

Reviewer's Responses to Questions

**Comments to the Authors:**

Reviewer #1: The authors addressed all my comments.

Reviewer #2: The authors have addressed my comments and have improved the quality and rigor of their manuscript. However, they still need to include some details in the main text. I don't think that responses that point to github scripts, i.e "The script (batch_process_results.py) provided in the GitHub provided the function of adjusting the setting of threshold 2 and generating the error rate of test dataset." are adequate. Please, include in the paper a simple description of how you choose the value of your parameter. The reader of a journal article is not supposed to parse a github script. The justification of using a larger step size of 91 should also be included in the main text.

Reviewer #3: The review is uploaded as an attachment.

**Have the authors made all data and (if applicable) computational code underlying the findings in their manuscript fully available?**

Reviewer #1: Yes

Reviewer #2: Yes

Reviewer #3: None

PLOS authors have the option to publish the peer review history of their article (what does this mean?). If published, this will include your full peer review and any attached files.

Reviewer #1: No

Reviewer #2: No

Reviewer #3: No

Figure Files:

Data Requirements:

Reproducibility:

References:

---

## [Decision Letter · Decision Letter 2]

30 Sep 2024

Dear Dr. Wu,

We are pleased to inform you that your manuscript 'DeepPL: a deep-learning-based tool for the prediction of bacteriophage lifecycle' has been provisionally accepted for publication in PLOS Computational Biology.

Best regards,

Yang Lu, Ph.D.

Academic Editor

PLOS Computational Biology

Stacey Finley

Section Editor

PLOS Computational Biology

Reviewer's Responses to Questions

**Comments to the Authors:**

Reviewer #1: The authors have addressed all my comments.

Reviewer #2: The authors have addressed my outstanding comments.

Reviewer #3: The authors have addressed all my questions, I suggest to accept this paper.

**Have the authors made all data and (if applicable) computational code underlying the findings in their manuscript fully available?**

Reviewer #1: None

Reviewer #2: None

Reviewer #3: None

PLOS authors have the option to publish the peer review history of their article (what does this mean?). If published, this will include your full peer review and any attached files.

Reviewer #1: No

Reviewer #2: No

Reviewer #3: No

---

## [Editor Report · Acceptance letter]

11 Oct 2024

PCOMPBIOL-D-24-00537R2 

DeepPL: a deep-learning-based tool for the prediction of bacteriophage lifecycle

Dear Dr Wu,

I am pleased to inform you that your manuscript has been formally accepted for publication in PLOS Computational Biology. Your manuscript is now with our production department and you will be notified of the publication date in due course.

With kind regards,

Anita Estes
